# Development of a Potentially Hazardous Pro-Glacial Lake in Aksay Valley, Kyrgyz Range, Northern Tien Shan

**Vitalii Zaginaev** [1,2,*], **Kristyna Falatkova** [2], **Bohumir Jansky** [2], **Miroslav Sobr** [2]  **and Sergey Erokhin** [1]

1   Institute of Water Problems and Hydropower, National Academy of Science, 720033 Bishkek, Kyrgyzstan; erochin@list.ru
2   Department of Physical Geography and Geoecology, Charles University, 128 43 Prague, Czechia; kristyna.falatkova@natur.cuni.cz (K.F.); bohumir.jansky@natur.cuni.cz (B.J.); sobr@natur.cuni.cz (M.S.)
*   Correspondence: zagivit@mail.ru; Tel.: +996-555-771434

**Abstract:** Debris flows caused by glacial lake outburst floods (GLOFs) are common hazards in mountain environments. The risk posed by glacial lake outburst hazards is particularly evaluated where the lower reaches of catchments are populated. A potentially dangerous lake has been identified adjacent to the Uchitel Glacier in Northern Tien Shan. This lake formed between 1988 and 1994 on the site of a retreated glacier in the upper part of the Aksay Valley. In this study we consider the possibility of an outburst of this pro-glacial lake in the future. The study involved bathymetry mapping of the lake, detailed profile sections of the valley, flow rate measurements on the Aksay river, and monitoring of the lake development using satellite images. Modelling of secondary debris flow inundation heights and hazard footprints has been undertaken. The outburst of this lake could cause powerful debris flows posing a threat to permanent residents living downstream, in the Ala-Archa Valley. Monitoring of the lake over the past ten years suggests certain changes in the runoff to the subsurface, and an increase in lake depth is observed. Glacial lakes with subsurface drainage are considered to be the most hazardous type as the knowledge of drainage channels functioning is still very limited and, thus, the timing of an outburst is hard to predict. Development of monitoring approaches to support forecasting of these hazards is of paramount importance to safety in mountain territories globally.

**Keywords:** pro-glacial lake; glacier retreat; Aksay Valley; Uchitel Glacier; GLOF; debris flow

---

## 1. Introduction

Climatic changes have a noteworthy impact on the cryosphere in high mountain regions, thereby affecting the magnitude and frequency of geomorphic processes such as glacial lake outburst floods (GLOFs), landslides, or snow avalanches [1,2]. GLOFs and debris flows are the most common natural hazards in mountain environments and are responsible for the large damage to infrastructure, and even loss of life. In Central Asia, climatic changes led to severe changes at high altitudes including intense permafrost degradation and glacier downwasting [3–5].

Natural hazards related to glaciers' degradation in high mountain terrains can have severe downstream consequences. GLOFs and glacier related debris flows have significant social and economic consequences in lower-elevation valleys and even in adjacent lowlands [6–9]. These phenomena have been described in all large mountain ranges around the world: in Central Asia [10,11], in the Swiss Alps [12,13], the Himalayas [14–16], and the Andes [17,18].

The Northern Tien Shan has very complex mountainous terrain and high tectonic and seismic activity. Active erosion processes are widespread on the slopes, with rock- and debris-slides and rockfall-type landslides. In narrow gorges, debris flow systems are common, often in connection to the degradation of glaciers and rapid formation of new hazardous lakes [11]. The northern slopes of the Kyrgyz Range differ in debris flow activity.

There are at least 199 potentially dangerous glacial lakes in Kyrgyzstan according to the catalog developed at Institute of Water Problems and Hydropower (Kyrgyz National Academy of Sciences). These lakes are of ice-cored moraine dammed type [19].

In the central part of the Kyrgyz Range, 22 cases of GLOF have been recorded since 1952 [19]. Additionally, the number and volume of moraine glacial lakes forming in the Ala-Archa Valley has increased [20,21].

The Aksay Valley is known for historic GLOFs and secondary debris flow events, and their genesis and frequency are well studied [22,23]. Repeated large-magnitude events have led to the formation of a large alluvial fan system at the mouth of the Aksay Valley [24]. The major events on record were in 1968 and 1969, triggered by an outburst of an englacial water pocket. During these events local authorities recorded peak discharge values of almost 900 and 800 $m^3 \ s^{-1}$, respectively [25].

Here, we combine several approaches, namely an assessment of a time series of aerial images, a detailed study of historical reports, a bathymetric survey of a pro-glacial lake, and valley profiles, to assess the potential outburst flood impacts. The object of study is a relatively small pro-glacial lake, which is in contact with the Uchitel Glacier situated in the Aksay Valley. The main objectives are to improve our knowledge of monitoring the development of dangerous lakes with subsurface drainage, using a combination of field work and remote sensing methods to evaluate possible impacts of lake' outburst floods and debris flows in populated areas.

## 2. Study Site

The Uchitel Glacier is located in the top part of the Aksay Valley (E 74°31′20″, N 42°31′10″), it has a NW orientation and its terminus lies at an elevation of 3630 m a.s.l. The Aksay Valley is located in the upper part of the Ala-Archa River basin (Figure 1a), which is part of the large endorheic Chu basin. The Aksay catchment size is 28.3 $km^2$, and its elevation ranges from 4895 m a.s.l. (Semenov Tianshanskiy peak) to 2200 m a.s.l. (lowest point of the fan). In the upper part of the catchment (above 3600 m a.s.l.), two valley glaciers can be found: the Uchitel Glacier (Figure 1d) and the Aksay Glacier (Figure 1c). The debris flow cone is a major landform situated at the valley mouth and it extends from 2350 to 2200 m a.s.l. The pro-glacial lake is at elevation of 3617 m a.s.l. The approximate volume of the lake is $27 \times 10^3 \ m^3$ (as of 2016) (Figure 1c,d).

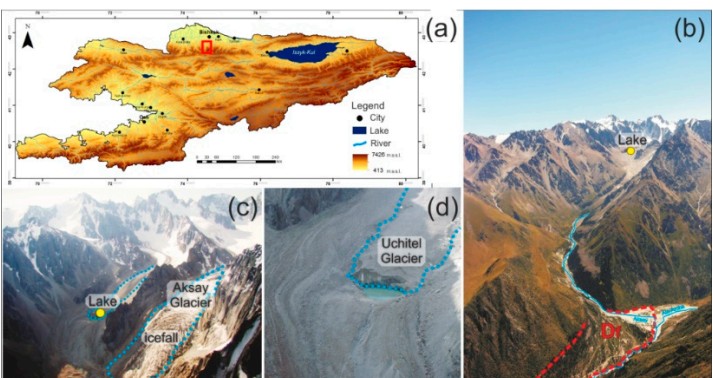

**Figure 1.** (**a**) Location of the Ala-Archa Valley (National Park) in Northern Tien Shan. (**b**) Oblique aerial photo overview of study site showing pro-glacial lake at the head of Aksay Valley, and the debris flow fan deposits in the low part of the Aksay Valley. (**c**) The Aksay Glacier and location of the lake adjacent to the Uchitel Glacier. (**d**) Close-up of the pro-glacial lake adjacent to the Uchitel Glacier.

In the lower part of the valley, a large debris cone has formed from a huge volume of debris flow deposits, approximately $11.6 \times 10^3$ m$^3$ [22,25]. Aggradation of the fan has pushed the course of the main Ala-Archa River to the western side of the valley. There is no resident population in the immediate Aksay Valley itself, however, the main access road into the national park traverses a large abandoned and vegetated part of the debris flow alluvial fan. The fan area offers flat ground and facilities for summer tourists visiting from Bishkek City and beyond, and with its car parking and amenities it is a popular base for climbers, trekkers, and picnickers. A single bridge 5 km down the valley provides a river crossing for the access road. The residence of the President of Kyrgyzstan and the administrative buildings of the national park are located 1 km from the central part of the cone.

The main morphometric characteristics for the Aksay Valley are given in Table 1. The high Melton index 0.6 [22] shows, that the Aksay Valley is a debris flow-dominated watershed and the average stream gradient calculated from archive 1:25,000 scale topographic maps is 0.17. The parameter of stream slope is very important and shows the ability of a valley to transform a flow process into debris flow processes [26].

**Table 1.** Main morphometric parameters of the Aksay Valley.

| | |
|---|---|
| Area catchment, km$^2$ | 28.3 |
| Watershed length, km | 10 |
| Highest point, m a.s.l. | 4895 |
| Lowest point (on the fan), m a.s.l. | 2200 |
| Melton ratio (Melton, 1965) | 0.6 |
| Mainstream length, km | 4.3 |
| Average stream gradient, m m$^{-1}$ | 0.17 |

The Aksay moraine-glacier complex is among the most dynamically developing glacier complexes in Kyrgyzstan. The top part of the complex is presented in Figure 1c. This complex provides all the natural conditions (geomorphological structure, melting of glacier) for formation of GLOF-induced debris flows in the future. These include glacier retreat, development of a lake, steep valley, and large source area of loose material accumulated in the valley floor. In contrast to other complexes in the Ala-Archa Valley, the Aksay complex creates a direct threat to infrastructure and people associated with rest areas and the road network which connects Bishkek to the national park. The Aksay River joins the main Ala-Archa River from the east (Figure 1b) and over the last 50 years it has modified the course of the main river substantially, pushing it towards the western valley slope. The active depositional lobe with levees currently directs the Aksay River flow towards the south, which is actually up the main valley.

*2.1. Geological Structure*

The Aksay Valley is situated in the higher, partly glaciated eastern part of the northern slope of the Kyrgyz Range (Northern Tien Shan Mountains). The Kyrgyz Range is 455 km long, trending roughly in the E-W direction, and reaches a maximal width of 40 km. This mountain belt belongs to a Caledonian fold zone and is composed mainly of granites and partially metamorphosed sedimentary rocks of Lower Paleozoic age [27]. About 60 km from Bishkek, the range reaches altitudes of 4600–4900 m. The Chu Basin, located on the northern site of the Kyrgyz Range, where the capital city of Bishkek is located, is filled by up to 5 km of Cenozoic deposits [28]. The lower part of the Ala-Archa Valley is dominated by Late Ordovician granitites, and in the higher parts, Silurian granites and granodiorites occur [29]. The middle to lower part of the Aksay Valley is formed by Upper Pleistocene deposits, and the uppermost part near the stream spring and the alluvial fan are formed by Holocene deposits. The steep canyon-like part of the valley and surrounding slopes are formed by granodiorites and granites (Late Ordovician complex). Slopes in the lower part of the valley are Early Ordovician diorites on the north side and mostly conglomerates on the south side. A thrust fault divides the rocks in upper and lower part of the valley.

## 2.2. Glaciers

The glaciated area of the Aksay catchment in 2017 was 8.5 km² [30]. The upper part of the valley contains two glacier bodies (Figure 2) separated by a bedrock ridge: the larger glacier, located on the western side, is called the Aksay Glacier, and the smaller one to the east is called Uchitel Glacier. The glaciers' tongues probably used to be connected in the past but due to their retreat since the Little Ice Age (LIA) is the most recent glacial event (approximately AD 1300–1850) signifies the cold period in the twentieth century [31,32] are the two difference glaciers. The Tien Shan glacier degradation since LIA is similar to changes observed in the Alps, Pamir, Alay, and Koryak plateau [33]. Differences between four regions in Tien Shan (northern, western, inner, and central) are small, though in the Northern Tien Shan the glacier retreat is more noticeable [34]. The tongue of the Uchitel Glacier is currently covered by debris and is undergoing thinning (Figure 2a–d). A pro-glacial lake, approximately 100 m in diameter, appeared between 1988 and 1994 (Figure 2c,d) at the terminus as the glacier melt water began to fill a depression in front of it. The Aksay Glacier has a heavily crevassed "icefall" in the lower part, the underlying disintegrating terminus is covered by debris as well.

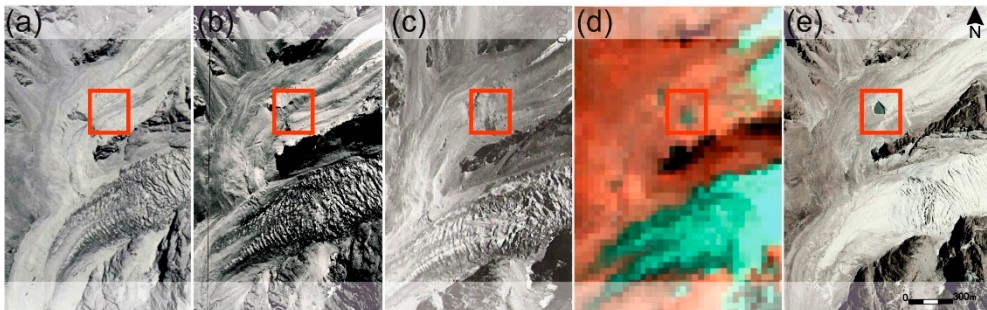

**Figure 2.** Archive aerial images of the Aksay and Uchitel Glacier tongues from (**a**) July 1960, (**b**) July 1978, and (**c**) August 1988, and satellite images: (**d**) Landsat image, July 1994, and (**e**) QuickBird image, August 2016. The red boundary is the location of the pro-glacial lake.

## 2.3. Climatic Conditions

According to a hydrometeorological station at Alplager (42°33′56.06″ N; 74°28′56.85″ E; 2150 m a.s.l.), located 1 km downstream from the end of the Aksay Valley, the mean annual precipitation in this region is about 560 mm. The maximum summer precipitation was recorded in June 1987, and reached 144 mm/month. The mean annual air temperature is 2.9 °C (Figure 3a), with mean maximum values of 27 °C in summer and the mean lowest values of −22 °C in winter. The maximum temperature on record was 31 °C (July 1983), and the lowest fell to −25.8 °C (December 2001). Figure 3 illustrates a long-term trend of increasing air temperatures and decreasing annual precipitation in the Ala-Archa Valley. The period with snow cover at the lower part of the valley usually spans from October until April. The average annual Aksay river discharge is 0.5 m³ s⁻¹. Summer time precipitation, which is almost solely of storm character, is also connected with strongly developed convection during powerful cold invasions [32]. In the winter months and in the first half of spring (March, April), precipitation is always solid and connected to the development of cyclonic indignations when cold invasions prevail. Since the end of April, the prevalence of storm rainfall events is observed. The early autumn season has the lowest precipitation, with rainfall only starting in the latter half of the season connected with the onset of cold invasions [35].

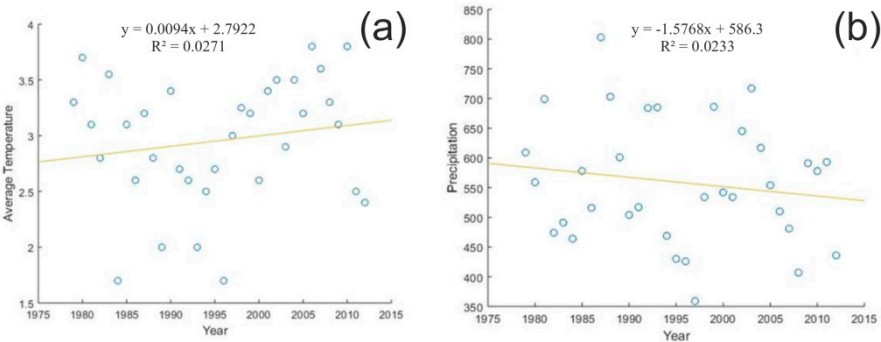

**Figure 3.** (**a**) Average annual air temperature (°C) and (**b**) annual precipitation totals (mm) from Alplager meteorological station in the period of 1979–2013.

## 3. Materials and Methods

Glacier retreat has been monitored since 1960 using available archive airborne and modern satellite imagery (1960, 1978, 1988, and a high-resolution QuickBird (Digital Globe, CO, USA) satellite image from 2016). The recent development (2012–2017) of the lake adjacent to the Uchitel Glacier was monitored based on satellite images from Google Earth® software. A 10 m spatial resolution DEM generated from Sentinel-1 image in SNAP® software was used to determine the catchment area boundary and to derive morphometric characteristics presented in the Table 1. The survey of the central and lower part of the valley was focused on detailed cross-section profiles. In total, 20 transverse and longitudinal profile (Figure S1) were surveyed in 2008 using a survey grade reflectorless total station (Leica TCR 705, Wetzlar, Germany). The longitudinal slope (gradient) of the channel bed (m m$^{-1}$) for the fans was calculated using data collected during field work in 2007 and 2015.

During the field reconnaissance of the debris flow path, we primarily focused on entrainment and accumulation processes in order to divide the debris flow path into different zones with similar geomorphic characteristics and to determine critical gradients for entrainment. The valley cross-sections were, therefore, examined in the field in order to estimate debris flow (max flood) height, i.e., the main parameter in the empirical relation given by Equation (1) [36], which combines the flow depth and channel slope. This parameter is very important in order to calculate which zones can be affected by debris flow processes in the future.

$$V = 4.83 \times H^{0.5} \times i^{0.25} \tag{1}$$

$$Q = A \times V \tag{2}$$

where $V$ is the peak velocity of the debris flow (m s$^{-1}$); $H$ is the debris flow depth determined at the cross sections (m); $i$ is longitudinal slope of the channel bed (m m$^{-1}$); $Q$ is discharge from the lake during/after an outburst flood (m$^3$ s$^{-1}$), and $A$ is cross-sectional area (m$^2$). For the calculations, we take the cross-sectional form of a rectangular area: $FA = HH \times BB$, where B is a width of a cross-section (m).

From the Equations (1) and (2) we calculate a value of debris flow depth (3):

$$H^{\frac{3}{2}} = \frac{Q}{4.83 \times B \times i^{0.25}} \tag{3}$$

Key characteristics in Equation (3) are discharge of debris flow, longitudinal slope and width of the channel bed. After transformation flow into debris flow expect a constant peak discharge. Based on cross-sections and parameter H, we can calculate a zone which will be affected by flooding.

Applying parameter H at the cross-sections, we can calculate which zones of the valley will be flooded under three scenarios. Based on historical data [37], we use the peak discharge for three realistic scenarios: 300 m$^3$ s$^{-1}$ for flows caused by partial outbursts, 600 m$^3$ s$^{-1}$ for major outbursts

from Aksay englacial water pocket (that were recorded in the past), 900 m$^3$ s$^{-1}$ for the most powerful flows (similar values estimated for events in 1968 and 1969 [22]).

The change in the position of the lake shoreline was measured with repeated surveys using a Leica TCR 705 total station [38]. To capture changes of the lake's depth and to calculate stored water volume, repeated bathymetric surveys were carried out in 2013 and 2016. Both surveys were carried out in August during the peak summer season as the lake volume is usually at its highest. An echo-sounder Lowrance Elite 5 (Tulsa, OK, USA) was used for the bathymetric measurements. The depth was recorded in a 2 m step at defined profiles, the collected data was interpolated (Kriging algorithm) to create a bathymetric map using ArcGIS 10.3® software.

A dam limiting the course of the Aksay stream on the debris fan was constructed in 2016 to reduce the risk from debris flows in the lower part of the valley. The structure and surrounding area was surveyed using photographs taken from a UAV, a point cloud was built in the Agisoft Photoscan 1.2.4®, St. Petersburg, Russia software and a DEM was created in ArcGIS 10.3®, ESRI, PR, USA. Ground-based and airborne survey data helped to fill in the data gaps in the DTM. The resulting DEM and valley cross-section profiles were used to calculate the areas potentially affected by debris flows (Figures S4 and S5).

For meteorological description were used an available data from Kyrgyz hydrometeorogical survey for last 50 years. Data of Aksay river discharges based on direct measurements, data is available for the last 15 years.

## 4. Results

### 4.1. Development of the Pro-Glacial Lake

The pro-glacial lake has formed at the terminus of the Uchitel Glacier in an actively developing intra-moraine depression. The depression formed during the phase of glacier retreat in last 60 years and still expands nowadays. The lake was formed between 1988 and 1994 and, thus, is considered to be a very young feature. The glacier is also falling into the lake from steep cliffs (up to 45–65°) which reach a height of 15–25 m. Meltwater streams from the glacier feed the lake and on the right flank of the glacier is the main meltwater channel of draining out from Uchitel Glacier. The lake extends back underneath the glacier (from 1994 to 2010), thereby contributing to more intensive warm-based melting (Figure 4). Figure 4d illustrates that during October 2015 (autumn) the lake level dropped rapidly from its position in August 2014 almost disappearing completely, but the lake had reformed again by August 2017. This behavior indicates that the drainage channels are located at the bottom of the lake and, more importantly, their capacity changes over time.

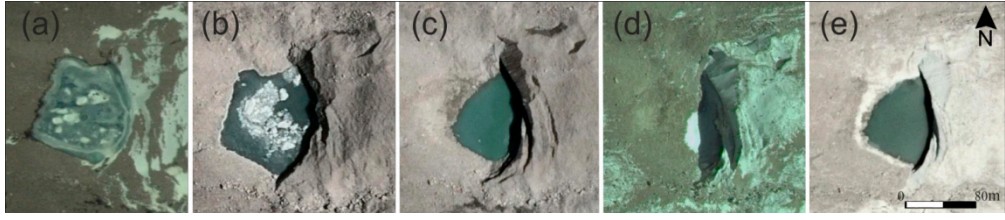

**Figure 4.** Satellite images from GoogleEarth®: (**a**) 27.05.2012, (**b**) 01.10.2013, (**c**) 30.08.2014, (**d**) 06.10.2015, and (**e**) 13.08.2017.

The observed change in the lake area over a 6-year period (2012–2018) is shown in Table 2. The area of the lake decreased between 2012 and 2015 (dropping from 6 733 m$^2$ to only 1603 m$^2$), but since then the lake has grown again, reaching an area of 4870 m$^2$.

**Table 2.** Changes in pro-glacial lake area between 2012 and 2018, based on satellite images from Google Earth.

| Satellite Image Date of Acquisition | Lake Area ($m^2$) | Lake Perimeter (m) |
|:---:|:---:|:---:|
| 27.05.2012 | 6733 | 322 |
| 12.07.2012 | 3760 | 239 |
| 22.04.2013 | 6597 | 316 |
| 01.10.2013 | 6559 | 326 |
| 30.08.2014 | 3716 | 236 |
| 06.10.2015 | 1603 | 172 |
| 08.06.2017 | 3347 | 239 |
| 13.08.2017 | 4180 | 261 |
| 01.08.2018 | 4870 | 271 |

During the 2010 field work, a scattered runoff (several flow paths) from the glacier surface was observed, supplying the lake with meltwater. The bed of the largest flow path, ran along the eastern margin of the glacier. The lake had a surface drainage running through the central part of a dam with water flowing through loose rocks without a clearly defined channel. A permanent channel was not incised probably due to low discharge of the flow and also due to stabilizing effect of a bedrock step (riegel) forming the dam. Figure 5a shows the results of the lake bathymetric surveys carried out in 2012 and 2016. The lake's area in July 2012 was $3.76 \times 10^3$ m$^2$ and the volume was calculated to $26.7 \times 10^3$ m$^3$ with a maximum depth of 12.1 m, and an average depth of 7.1 m.

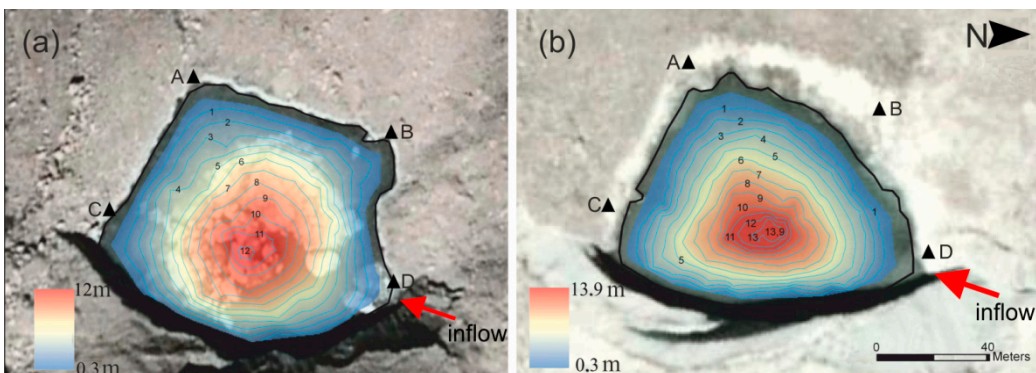

**Figure 5.** (**a**) Bathymetry map of the pro-glacial lake adjacent to the Uchitel Glacier in July 2012 (**b**) and in August 2016. The letters A–D refer to reference points used for the survey (3630 m a.s.l.).

In 2016, the lake level reduced to 6 m below the dam crest (Figure S3). The lake's area slightly decreased from 3800 m$^2$ to 3500 m$^2$, however, the depth of the basin increased from 12 m (15 June 2012) to 13.9 m (20 August 2018) (Figure 5b). This indicates melting of the buried ice at the lake bottom and lowering of the lake floor. The total volume of the retained water remained almost unchanged between 2010 ($26.7 \times 10^3$ m$^3$) and 2016 ($27.1 \times 10^3$ m$^3$). The lake basin subsidence led to a change in the character of the runoff from the lake: switching from predominantly surface overflow to flow via subsurface drainage channels. This change in drainage characteristics is important as it increases the lake's outburst potential.

Currently, the lake is still drained by subsurface channels. Usually the sources of the channels of underground flows appear on the sides of pro-glacial lakes in the form of thermokarst craters [22]. On the sides of the studied pro-glacial lake, however, such thermokarst craters have not appeared yet. Therefore, it is difficult to identify the exact point where the underground runoff from the lake begins. The main resurgence of the seepage water from the lake is located 500 m downstream at an elevation of 3493 m a.s.l. The water discharge at the time of the survey in 2016 was 0.06 m$^3$ s$^{-1}$, which is relatively low. Part of the water very likely stays underground, so the real runoff is probably higher.

The lake at the Uchitel Glacier is a non-stationary one with dynamic changes in its area and volume, depending on the glacier retreat and melting of buried ice at lake's bottom. Additionally, the observed fluctuations in the lake water level are significant, indicating the drainage channels have limited capacity. Taking into account the lake's changing subsurface drainage and the fact that the lake has a potential to grow and increase its volume, it was categorized as a potentially dangerous lake.

### 4.2. Transport of Debris Flow Material

Sources of the transported material are erosion of the bedrock by the stream, glacial till left on valley floor after glacier retreat, and material from hillslope rock weathering that was transported to the valley floor by gravitational force (landslides, rockfall). The slopes are steep especially in the upper part—near the canyon-like channel it reaches 42°, somewhere even over 50°, near the confluence with a tributary it lowers to around 36°. The slopes in the lowest part of the valley (just above the fan) are between 21° and 27°.

This stream flows through a steep canyon-like valley for 1100 m before the valley floor becomes less steep and more open. This section is followed by a rather flat section (1200 m) where deposition of transported material takes place. The lower part of the channel is characterized by several overlapping facies—deposits of lateral accretion forming elongated shapes. These are remnants of hyperconcentrated flows and debris flows that occurred in the Aksay Valley [37]. They were triggered either by intensive precipitation or sudden glacier meltwater release. The analysis of satellite images aimed to capture key morphological features and distinguish sedimentary formations of different material (color, grain size) within the channel.

On the alluvial fan (Figure 6), three types of morphologies emerged: (a) lobate crevasse-splay deposits, (b) levee deposits, and (c) former braid bars. The crevasse splay deposits are fan-shaped wedges of coarse sediment deposited downstream of levee breaks during the peak discharge. The next distinctive landforms are levee deposits—long linear ridges of sediment that form adjacent to the channel. The most abundant formations are bars of bouldery material—these are typically situated on the upper part of the fan where the slope gradient is reduced, the stream widens, and the stream's transport capability is significantly reduced.

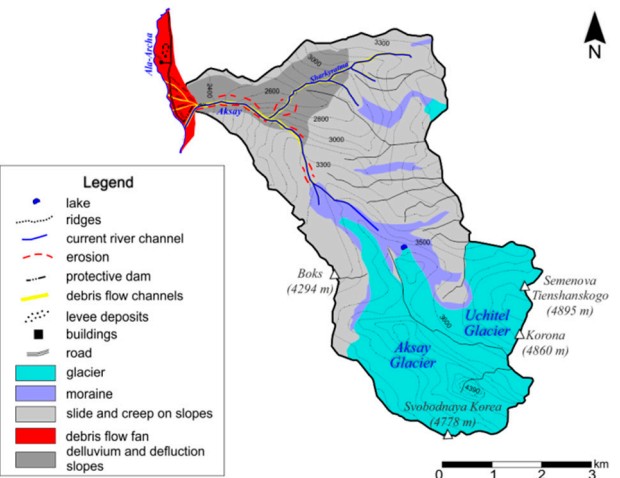

**Figure 6.** Geomorphologic map of the Aksay Basin.

### 4.3. Threat from the Aksay Glacier

In the past, the Uchitel and the Aksay Glaciers converged. We assume that this is was during the LIA. Since the 1960s, there are two distinct glaciers visible in the aerial/satellite images (Figure 2a). After 1960, the Aksay Glacier tongue has experienced rapid downwasting (Figure S2) with an average reduction in mean surface elevation by almost 25 m and maxima in the lower parts of the snout of up to 60 m [20]. All reported GLOFs in the valley were associated with the activity in the Aksay

Glacier [22], particularly with outburst from an englacial water pocket. In the past 50 years, no such outburst was reported, but there is an emerging threat from the development of lakes in the lowest part of the Aksay Glacier.

The development of the lake depression in front of the Aksay Glacier margin is influenced primarily by the glacier movement and the seasonal changes in its balance. Below the icefall, the tongue turns in the northwest direction, gradually disappearing under a layer of talus and its edge ends at an altitude of 3320 m a.s.l. The location of the edge of the glacier in comparison with most glaciers in the other studied sites [30] is significantly lower. In the period 1978–1988 the glacier was stationary and began to recede only during the 1990s. The total distance of the linear retreat of the glacier did not exceed 330 m in the entire study period. The relatively slow retreat of the Aksay Glacier is caused by the insulating effect of the debris cover on its surface and its location in the shadow of the north-facing valley walls.

The repeated formation of the lake takes place in the area of the degrading debris-covered glacial snout. Here, the orientation of the valley changes from NW-SE to N-S. The glacier's terminus is currently (2018) located at an altitude of 3320 m a.s.l., following up on a transition into the morphologically complex accumulation relief of the valley bottom. Depressions at the Aksay Glacier terminus appeared in connection with the degradation of stagnant ice and with the action of glacial melt waters. The result was a strongly dissected relief over the entire width of the valley bottom, which is characterized by alternating elevations and depressions of irregular shapes. From the point of view of the area, the largest depression (Figure 7) coincides with the active glacier terminus, but it is mainly developed in the surrounding glacial accumulations. In the down valley direction the depression is blocked by a shaft 5–7 m high, which was separated from the stationary glacier during the regression in the 1990s. In 2004, the depression was filled with water but in the following two years meltwater drained through an assumed system of subsurface canals. This development of a lake depression is characteristic of thermokarst lakes in which water gradually accumulates and then flows from the lake into underground.

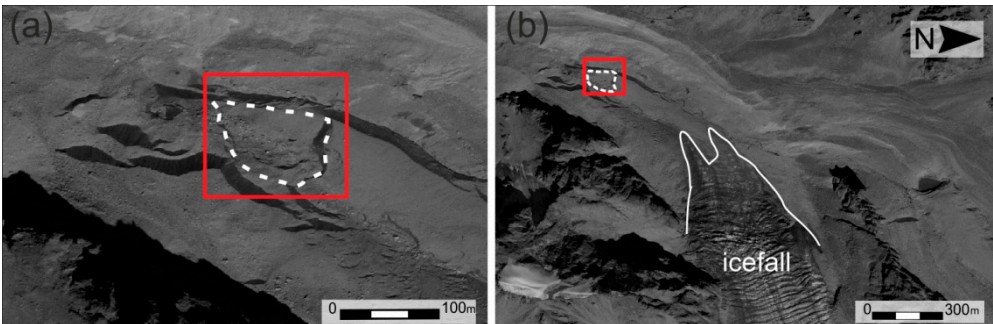

**Figure 7.** A depression formed at the Aksay Glacier terminus (Worldview-3, 2014). (**a**) Front part of the Aksay Glacier with development of ground surface depressions; (**b**) overview of the Aksay Glacier terminus (white line).

At the moment, the depressions in front of the Aksay Glacier are empty and all the water coming from the glacier bypasses them. However, based on previous experience it is possible that this situation can change very quickly.

### 4.4. An Area Potentially Affected by a Debris Flow

Three scenarios of the possible debris flow magnitudes are considered, the size variation based on archival information on outbursts events in the Aksay Valley [22,24,25,37] and incorporating data about recent outbursts of similar lakes located in the neighboring Teztor Valley [26,39]. Three peak flow discharge rates options for the future were considered: 300, 600 and 900 $m^3$ $s^{-1}$. These are peak discharges of a debris flow that moves down the valley (i.e., water and entrained sediment and

boulders), the discharge of the water flow from outburst itself would be much lower. Moving down the valley, the debris flow discharge height increases due to narrowing of the drainage channel.

The upper part of the Aksay fan is most vulnerable to debris flows, where the valley turns into a wide valley from a narrow gorge. In this part of the fan there is an artificial protective dam that was constructed in 2016. Its length is marked by a dotted black line in Figures 6 and 8b. The average height of the dam built is 0.80 m from the river bed and at the most critical place in the upper part of the active fan cone, indicated in profile 4 (P4 in Figure 8), the height of the dam is 0.75 m. At a debris flow discharge of 300 $m^3$ $s^{-1}$ the flow height at this point will be 0.8 m, thus breaching the dam (constructed from big stones, diameters, which is placed to divert the flow to the south. Furthermore, at the higher flow rates of 600 and 900 $m^3$ $s^{-1}$, the flood height is modelled to be 1.1 and 1.4 m, respectively. Moving further down into the flatter and wider part of the valley, the flow height decreases, and deposition in the apron zone is dominated by finer grained mud and silt deposits. At a flow rate of 300 $m^3$ $s^{-1}$ in the narrowest sections of the valley, the maximum flow heights were 2.2 m (P18) and 2.9 m (P14). The highest flood height will be for the scenario with a flow rate of 900 $m^3$ $s^{-1}$ and will be 6.3 m at the initiation point (P20) where the canyon-like valley is very narrow. Figure 8c indicates the land area that can be potentially affected by a debris flow according to the three scenarios and based on the input parameters given in Table 3. The cross-sections were created in MATLAB®, MathWorks and the heights extrapolated in ArcGIS 10.3®, ESRI software to draw the debris flow hazard footprint maps.

**Table 3.** Debris flow depth along the cross-section profiles in the Aksay Valley for the three scenarios.

| Cross-Section | H (m) | | |
|---|---|---|---|
| | $Q_{300}$ | $Q_{600}$ | $Q_{900}$ |
| P 20 | 3.4 | 5 | 6.3 |
| P 19 | 1.9 | 2.8 | 3.6 |
| P 18 | 2.2 | 3.2 | 4.1 |
| P 17 | 1.6 | 2.3 | 2.8 |
| P 16 | 2.1 | 3 | 3.8 |
| P 15 | 1.9 | 2.8 | 3.5 |
| P 14 | 2.9 | 4.2 | 5.3 |
| P 13 | 2 | 3 | 3.8 |
| P 12 | 1.9 | 2.8 | 3.6 |
| P 11 | 1.6 | 2.3 | 2.9 |
| P 10 | 1.2 | 1.7 | 2.2 |
| P 9 | 1.2 | 1.7 | 2.1 |
| P 8 | 1.1 | 1.6 | 2 |
| P 7 | 0.9 | 1.4 | 1.7 |
| P 6 | 0.9 | 1.3 | 1.7 |
| P 5 | 0.8 | 1.2 | 1.5 |
| P 4 | 0.8 | 1.1 | 1.4 |
| P 3 | 0.6 | 0.8 | 1.1 |
| P 2 | 0.4 | 0.6 | 0.9 |
| P 1 | 0.3 | 0.6 | 0.8 |

According to the scenario (300 $m^3$ $s^{-1}$), the artificial dam will be overflown in the upper part (P4) but probably not destroyed completely, as the modelled inundated area is limited to the southern part of the fan (surrounding the current channel location). There are not many human activities anyway. In case of the scenario of 600 $m^3$ $s^{-1}$, the situation would be significantly different. A large part of the fan would be affected by the debris flow as it would breach the artificial dam and continue directly to the opposite valley slope. In case of the third scenario, the dam can be destroyed and in the central part of the fan the Ala Archa River could be dammed. It is a most negative scenario.

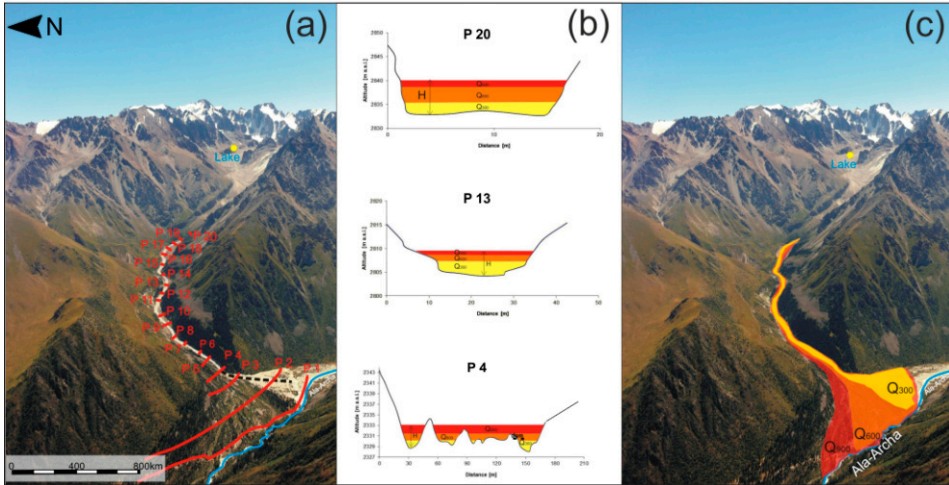

**Figure 8.** (**a**) Location of cross-section profiles P1–P20 across the Aksay Valley. (**b**) Selected representative cross-sections in the upper, middle, and lower part of the valley, showing modelled water height for each debris flow scenario: 300 (yellow), 600 (orange), and 900 m$^3$ s$^{-1}$ (red). (**c**) Potentially affected zones during a debris flow event according to the given scenario.

## 5. Discussion

The Aksay Valley is one of the most potentially dangerous places in the Ala-Archa National Park. The number of visitors to the valley is constantly increasing with around 500 visitors per day during summer months (June–August) and debris flows and GLOFs pose a threat to them. In the case of the large-scale debris flow scenario (peak discharge of 900 m$^3$ s$^{-1}$), the infrastructure of a large part of the valley would be threatened.

The main conditions on which the formation of debris flows depend are: (a) climatic and microclimatic conditions of the region (especially the distribution of precipitation, its formation, snow accumulation pattern, glacier melting characteristics); (b) geomorphological conditions of the valley (size and shape of the watershed, its altitude, the slopes and the structure of the valleys of mountain rivers); (c) geological conditions that determine the accumulation and volume of loose material (development of surface processes, such as weathering and gravity processes, rock/material type and grain size, ancient and neo-tectonic movements along faults, and seismicity).

Since the end of the LIA, the glaciers of Central Asia and other mid-latitude regions have tended to retreat rapidly. The number of potentially dangerous lakes increases every year [19]. Additionally, the size of existing lakes changes [20]. The pro-glacial lake adjacent to the Uchitel Glacier formed in a depression in a moraine-glacier complex after the glacier retreated. This development is reflected, first of all, in the deepening of a depression, as shown by the bathymetric survey measurements of 2010 and 2016. Until 2010, the lake level was at the top edge of the moraine dam, and the lake had an overspill and a surface runoff regime. The depression is still very young (began to form after 1988); it grows with phases of glacier retreat, and will likely further develop in the future.

A common trigger for the formation of a large magnitude debris flow is a glacial lake outburst flood. In our case, the triggers include accumulated meltwater in the depression of the Aksay moraine-glacier complex (the stagnant buried terminus), and the pro-glacial lake, which can burst due to increased volume (and built-up hydrostatic pressure) and limited subsurface channels capacity. The volume of these water reservoirs is the main indicator of its outburst potential, and thus the potential of secondary hazards such as debris flow in the Aksay Valley. Other forms of accumulation of glacial meltwater must not be omitted, such as englacial water pockets of the Aksay Glacier. Outbursts from the Aksay Glacier were the historic cause of major debris flows and the reason behind recent growth and progradation of the Aksay alluvial fans in the 1960s [37]. Due to the contraction of the Aksay Glacier during its retreat, the storage capacity of englacial water pockets was

probably significantly decreased, but did not disappear completely, as they are probably linked to the existing icefall.

Currently, perhaps the most dangerous scenario for Aksay Valley is the situation in which a moraine-glacier complex accumulates water in a lake adjacent to the Uchitel Glacier. The dam of this lake consists of rock outcrop covered with moraine deposits containing ice, and its erosion and an outburst is possible when the lake drainage changes to superficial. Due to the fact that the lake has subsurface drainage, the fluctuations in the water level directly depend on the capacity of the subsurface drainage channels. When the lake is filled up and overflows, there may be additional pressures on the drainage channels so that they can open. Meanwhile, a current volume of lake adjacent to the Uchitel Glacier is too small and 600 m$^3$/s and 900 m$^3$/s scenarios is unrealistic but in the future when volume of lake can increase and can activate an Aksay Glacier it is a possible a formation of a powerful events.

An outburst flow, encountering on its way a deep and steep Aksay Valley with large volumes of loose morainic and landslide deposits, is expected to be transformed into a debris flow. Its flow can be calculated as in events in the past (the largest recorded flow was 900 m$^3$ s$^{-1}$ during the 1968 event [22]). Such a debris flow is capable of restructuring the entire Aksay fan morphology, destroying the protective artificial dam at the top part of the cone, as well as a rest area and may even block the Ala-Archa River. In the worst-case scenario, the main Ala-Archa River could be blocked by the deposited sediment. As a consequence, a lake may form behind this temporary dam and when breaching, it may threaten the settlements in the lower Ala-Archa Valley, but also the capital, Bishkek.

Additionally, we cannot exclude the option of a debris flow formation after intensive rains. The torrential rains in the Aksay river valley historically caused debris flows from the right tributary of the Aksay Sharkyratma River in 1999 and 2003, when rainfall intensity exceeded a significant trigger value from 10 to 30 mm per hour for a duration of at least 3 to 1 h, respectively (according to the weather station Alplager). The valley is especially hazardous during heavy rainstorms, often occurring during May–June, when avalanche snow accumulates in the bottoms of the deep canyons of Aksay and Sharkyratma Valleys. This wet avalanche snow meltout is an additional source of water supply contributing to debris flows, which significantly increases their flow rate and power.

To manage the hazards from GLOF and debris flows in the Aksay Valley it is necessary to continuously monitor lake levels and glacier condition to provide reliable data and timely interpretation to make informed decisions. It is also necessary to develop an appropriate early-warning system and demarcation of safe zones for evacuations. It is recommended, based on our modelling, that the height of the protective dam on the fan is increased by at least 1 m, since at even the lowest flowrate of 300 m$^3$ s$^{-1}$ it will be destroyed. In the worst case scenario there is a threat of blocking of the Ala-Archa River bed by the deposited debris flow, how it was after events in 1968 and 1969 [37] debris flow blocked an Ala-Archa River after that was a secondary flood. Debris flow damaged of roads, bridges, exposed buildings, and other infrastructure [22].

## 6. Conclusions

Over the past eight years, the pro-glacial lake in the Aksay Valley has changed its drainage regime from surface to subsurface. The depth of the lake increased from 2013 to 2016 to 13.9 m, which indicates melting of the ground-based ice/sediment making up the bottom of the lake. The most likely option is the water will break through via subsurface flow channels. Experience from historic GLOF events that took place across the Tien Shan mountains [26,39] show that even a lake with small volume can cause a debris flow with a flow rate of 300 m$^3$ s$^{-1}$. In case of further glacier retreat, an increase in the volume of the pro-glacial lake adjacent to the Uchitel Glacier is possible and then there is the potential for even more powerful flows than those that have occurred in recorded past [37].

Our study shows that even with small flows (300 m$^3$ s$^{-1}$), a protective artificial dam built to protect the valley may be easily overflown and prove ineffective and its buildup and strengthening is necessary. Continuous monitoring of the pro-glacial lake and surrounds is needed, including

further field investigation (i.e., geophysical survey and monitoring), ground based and hydrological monitoring and research, and the application of Remote Sensing methods.

**Supplementary Materials:** The following are available online http://www.mdpi.com/2306-5338/6/1/3/s1.

**Author Contributions:** Z.V. and E.S. made a bathymetry survey on the lake, geomorpological mapping, analysis of satellite images and archival aerial images and reports. J.B. and S.M. conducted the field work and did a cross-sections in the valley. F.K. and Z.V. discussed and contributed to the writing of the manuscript.

**Funding:** This research has been funded by the project "International Mobility of Researchers at Charles University" reg.no. CZ.02.2.69/0.0/0.0/16_027/0008495.

**Acknowledgments:** This work has been supported by the project "International Mobility of Researchers at Charles University" reg.no. CZ.02.2.69/0.0/0.0/16_027/0008495.

**Conflicts of Interest:** The authors declare no conflict of interest.

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
