# Peer review of "Development of a Potentially Hazardous Pro-Glacial Lake in Aksay Valley, Kyrgyz Range, Northern Tien Shan"

_hydrology, doi:10.3390/hydrology6010003_

Round 1
Reviewer 1 Report
General Comments
This is an excellent paper that looks at the development of a potentially hazardous pro-glacial lake in the Tien Shan. Whilst the details of the study are essentially a local study, the implications of the work has much wider significance. There are a few minor points that need addressing, including increasing the scope of the references to the Little Ice Age and glacier evolution in the Tien Shan, with comparisons with other mid-latitude mountain areas.
Specific Comments
1. In section 2.2 Glaciers, the Little Ice Age needs some brief description. Please define the LIA here and refer to other examples from the mid-latitudes where LIA glaciers were bigger than today. Lots of examples from Europe and the Mediterranean. The Mediterranean mountains are probably most relevant here because they are at a similar latitude at c. 30-45°N. References and definitions of LIA can be found in these review papers:
Hughes, P.D., 2018. Little Ice Age Glaciers and Climate in the Mediterranean Mountains: A New Analysis. Cuadernos de Investigación Geográfica 44, 15-46. https://publicaciones.unirioja.es/ojs/index.php/cig/article/view/3362
Hughes, P.D. 2014. Little Ice Age glaciers in the Mediterranean mountains. Mediterranée 122, 63-79. https://journals.openedition.org/mediterranee/7146
In addition to Aizen et al’s paper, are there any other LIA papers from the region, between the Mediterranean and the Tien Shan? What about the Caucasus? This point is important because retreat of glaciers since the LIA has direct implications for the development of hazardous glacial lakes in the mid -latitude mountains.
A brief search revealed some other LIA papers that are relevant from the Tien Shan, such as:
Yanan Li et al. 2017. A Review on the Little Ice Age and Factors to Glacier Changes in the Tian Shan, Central Asia. https://www.intechopen.com/books/glacier-evolution-in-a-changing-world/a-review-on-the-little-ice-age-and-factors-to-glacier-changes-in-the-tian-shan-central-asia
Li et al. 2016. Cosmogenic 10Be constraints on Little Ice Age glacial advances in the eastern Tian Shan, China. Quaternary Science Reviews Volume 138, 15 April 2016, Pages 105-118.
2. RESULTS. Add word [The] to lake
4.1. Development of [the] lake
[The] lake was formed in the end part of Uchitel glacier
Change “The glacier falls off a steep cliffs” to “The glacier falls off steep cliffs”
There are other areas where you refer to “lake”. Why not call it “pro-glacial Aksay lake” like in the title? You only use that name in the title but not elsewhere in the manuscript.
3. DISCUSSION. You need to expand the discussion to highlight why this research is important for other mid-latitude mountain areas where glaciers have retreated since the Little Ice Age and especially were glacier retreat is currently rapid. See references provided earlier.
Author Response
Reviewer #1
General Comments
This is an excellent paper that looks at the development of a potentially hazardous pro-glacial lake in the Tien Shan. Whilst the details of the study are essentially a local study, the implications of the work has much wider significance. There are a few minor points that need addressing, including increasing the scope of the references to the Little Ice Age and glacier evolution in the Tien Shan, with comparisons with other mid-latitude mountain areas.
Response: We thank the reviewer for the comments and suggestions which helped to improve our work.
Specific Comments
1. In section 2.2 Glaciers, the Little Ice Age needs some brief description. Please define the LIA here and refer to other examples from the mid-latitudes where LIA glaciers were bigger than today. Lots of examples from Europe and the Mediterranean. The Mediterranean mountains are probably most relevant here because they are at a similar latitude at c. 30-45°N. References and definitions of LIA can be found in these review papers:
Response: Thank you for the comments and references. We agree that it would be good to add a brief description of LIA, we incorporated it in the text with relevant references.
In addition to Aizen et al’s paper, are there any other LIA papers from the region, between the Mediterranean and the Tien Shan? What about the Caucasus? This point is important because retreat of glaciers since the LIA has direct implications for the development of hazardous glacial lakes in the mid -latitude mountains.
Response: In this paper - Stokes, et. al 2007, our colleagues from Moscow State University estimated development of pro-glacial lakes in Caucasus. We collaborate closely with glaciologists and limnologists from Russia, China, and Switzerland. The topic of development of hazardous lakes is very important.
Stokes, C. R., Popovnin, V., Aleynikov, A., Gurney, S. D., & Shahgedanova, M. (2007). Recent glacier retreat in the Caucasus Mountains, Russia, and associated increase in supraglacial debris cover and supra-/proglacial lake development. Annals of Glaciology, 46, 195-203.
A brief search revealed some other LIA papers that are relevant from the Tien Shan, such as:
Yanan Li et al. 2017. A Review on the Little Ice Age and Factors to Glacier Changes in the Tian Shan, Central Asia. https://www.intechopen.com/books/glacier-evolution-in-a-changing-world/a-review-on-the-little-ice-age-and-factors-to-glacier-changes-in-the-tian-shan-central-asia
Li et al. 2016. Cosmogenic 10Be constraints on Little Ice Age glacial advances in the eastern Tian Shan, China. Quaternary Science Reviews Volume 138, 15 April 2016, Pages 105-118.
2. RESULTS. Add word [The] to lake
4.1. Development of [the] lake
[The] lake was formed in the end part of Uchitel glacier
Change “The glacier falls off a steep cliffs” to “The glacier falls off steep cliffs”
There are other areas where you refer to “lake”. Why not call it “pro-glacial Aksay lake” like in the title? You only use that name in the title but not elsewhere in the manuscript.
Response: We modified the title to avoid a misunderstanding. In this valley, there is the Aksay Glacier and the Uchitel Glacier. In the past, the Aksay Glacier was more active and all GLOF events were linked to this glacier.
3. DISCUSSION. You need to expand the discussion to highlight why this research is important for other mid-latitude mountain areas where glaciers have retreated since the Little Ice Age and especially were glacier retreat is currently rapid. See references provided earlier.
Response: Thank you for the comment. We added information to the discussion section and also we added conclusions.

Reviewer 2 Report
General comments:
Zaginaev et al. aim to address the formation of proglacial ‘lake’ in the Aksay valley in norther Kyrgyzstan and assess the potential for a glacial lake outburst flood. The study is interesting, but there are currently key methodological details missing, and statements that this lake is very hazardous are currently not supported by the analysis. The study therefore requires further work before publication.
1. The introduction and discussion sections are very scarce on literature. There are many papers dealing with GLOFs in other mountain ranges such the Peruvian Andes and across the Himalaya. Some of these studies should be discussed in order to put your study into context. The ‘lake’ is actually better termed a ‘pond’ due to its small size (<20,000 m2). This means it should probably be discussed as a proglacial pond or water body. The small size makes the lake very distinct from other studies dealing with GLOFs that investigate lakes of a much greater size, e.g. Rounce et al. (2017a). Pond or small lake outbursts can occur, e.g. Rounce et al. (2017b) and Miles et al. (2018), but the damages are usually limited. Since the pond is very small in your study, it is hard to image it would form a damaging debris flow, but limited socio-economic details are presented.
2. There are no details how the debris flow simulation was carried out. What debris flow modelling software was used? What was the outburst hydrograph? What resolution was the DEM? Did the model inundate trekking paths? How long did it take for the lake volume to drain?
3. There is no conclusion. Please add one.
4. The pond has drained during your study e.g. Figure 4d, so why is this drainage, which I assume did not cause a debris flow, likely to become a dangerous event? The volume is very small, so it would only be able to sustain the peak discharge for a very short time.
Specific comments:
L9. Remove ‘The’.
L14. Perhaps ‘failure of the moraine dam’, or ‘outburst of the proglacial lake’, rather than failure of the lake.
L14-15. Fieldwork and monitoring are nonspecific. State what was done in the field and monitored.
L17. State how many people. State specific years.
L26. I don’t think it’s true to say lakes with conduits are most hazardous.
L33. Please rephrase ‘tectonically active slopes of ranges are very dynamic.’
L34-35. Add citations to support these statements.
L38. GLOFs
L48. Historical archives of what? Please state.
L60. Figure 1. A, should probably be Figure 1a. Check the journal formatting and correct throughout.
L64. Add ‘Elevation (m a.s.l)’ to the panel (a) legend.
L83. The catchment area was already stated earlier. Define the ‘high Melton index’.
L94. Conditions such as what? Please state.
L114. Please specify the date.
L120. In the abstract it is stated after 1980. You need to be specific about the formation and development dates. Add more details in supplementary information if required.
L122. Where is the icefall, please label.
L125. Figure 2 doesn’t show when the lake appeared, it could be anywhere from 1988 to 2016. Can you add a Landsat image and be specific about the formation date. State the specific dates for each image.
L160. Add the equation and r2 value to the trendlines.
L164. What about Landsat imagery?
L168. Where did this 10 m DEM come from, how was it generated?
L207-208. State the model and uncertainty in the echo sounder used. State what algorithm did you use to interpolate the bathymetry and why?
L211. State Agisoft Photoscan and add the version number. How was the DEM built in Agisoft? Did you use ground control points, or the native UAV georeferencing. What was the RMSE?
L213. There are currently no details on the modelling.
L214. ‘digital elevation model’. How was the digital model combined with the profiles?
L217. ‘The lake…’
L218. How young is very young?
L219. You need to be more specific about the formation date.
L220. ‘falls off’ is not a good term. It is either flowing or calving. Use the degree sign, not ‘0’
L221. These are commonly referred to as ‘ice cliffs’ e.g. Watson et al. (2017b), .
L223. ‘Figure 4d’.
L238. ‘field work’. Annotate Figure 5 to show where meltwater is entering the lake and ‘the dam’. Replace 0,3 with 0.3 on the legend, and 13,9. Why does the scale bar suggest the lake is approximately 320 m adjacent to the ice cliff in Figure 5, but ~100 m in Figure 4? State the reference elevation of the bathymetric surveys, other the two surveys cannot be compared. The shape of the surveys looks different, suggesting a different water level. So specify the elevations so you can be sure the basin has deepened.
L245. Add the July 2012 lake area to Table 2.
L254. State the dates of this area decrease.
L264. Capitalise ‘Uchitel Glacier’ here and throughout. Similar for ‘Uchitel Lake’.
L271. State in cumecs no litres.
L272. The lake is adjacent to the glacier, not ‘under’, which would imply a subglacial lake. Change throughout.
L276. ‘Very hazardous’ isn’t supported by this study without evaluating the mechanisms of how the lake would outburst. Especially if there currently isn’t any water in it. The lake has a very small volume, comparable to supraglacial ponds on Himalayan debris-covered glaciers which drain and reform annually e.g. Miles et al. (2017). Outburst from these ponds can occur, but the volumes are typically much larger, especially to cause any notable impact. See Rounce et al. (2017b).
L338. (Figure 7). Please add the glacier outlines to this figure, and perhaps the lake outline. It is not clear otherwise.
L350. What is the ‘end wall’?
L367. Remove –.
L369. There is a difference between the lake benignly draining through conduits and outbursting.
L382. Figure 8
L380-399. There are no details in the methods of how this simulation was carried out. What software? Was the input hydrograph? How was the debris entrainment determined?
L404 Table 3. Replace commas with ‘.’
L407. National Park.
L409. If you are stating a GLOF is a danger, you should indicate a probability of occurrence. How many visitors visit annually?
L410. Does your simulation reach this city? It is not shown. Where would the large GLOF event originate given the lake is small?
L427. Be specify about the lake formation.
L445. Underground is probably incorrect. I assume you mean through ice, or englacial.
L452. Restructuring.
L454-457. This is just speculation and should be removed or supported with evidence from your study.
L470. Evacuation of who? The trekkers?
References
Miles, E.S. Steiner, J. Willis, I. Buri, P. Immerzeel, W.W. Chesnokova, A. and Pellicciotti, F. 2017. Pond Dynamics and Supraglacial-Englacial Connectivity on Debris-Covered Lirung Glacier, Nepal. Frontiers in Earth Science. 5(69). http://dx.doi.org/10.3389/feart.2017.00069.
Rounce, D.R. Byers, A.C. Byers, E.A. and McKinney, D.C. 2017b. Brief communication: Observations of a glacier outburst flood from Lhotse Glacier, Everest area, Nepal. The Cryosphere. 11(1), 443-449. 10.5194/tc-11-443-2017.
Watson, C.S. Quincey, D.J. Carrivick, J.L. Smith, M.W. Rowan, A.V. and Richardson, R. 2017a. Heterogeneous water storage and thermal regime of supraglacial ponds on debris-covered glaciers. Earth Surface Processes and Landforms. 229-241. http://dx.doi.org/10.1002/esp.4236.
Watson, C.S. Quincey, D.J. Smith, M.W. Carrivick, J.L. Rowan, A.V. and James, M. 2017b. Quantifying ice cliff evolution with multi-temporal point clouds on the debris-covered Khumbu Glacier, Nepal. Journal of Glaciology. 823-837. http://dx.doi.org/10.1017/jog.2017.47.
Author Response
Reviewer #2
General comments:
Zaginaev et al. aim to address the formation of proglacial ‘lake’ in the Aksay valley in norther Kyrgyzstan and assess the potential for a glacial lake outburst flood. The study is interesting, but there are currently key methodological details missing, and statements that this lake is very hazardous are currently not supported by the analysis. The study therefore requires further work before publication.
Response: We thank the reviewer for the time and effort dedicated to the revision of our work, which significantly improved the original version.
1. The introduction and discussion sections are very scarce on literature. There are many papers dealing with GLOFs in other mountain ranges such the Peruvian Andes and across the Himalaya. Some of these studies should be discussed in order to put your study into context. The ‘lake’ is actually better termed a ‘pond’ due to its small size (<20,000 m2). This means it should probably be discussed as a proglacial pond or water body. The small size makes the lake very distinct from other studies dealing with GLOFs that investigate lakes of a much greater size, e.g. Rounce et al. (2017a). Pond or small lake outbursts can occur, e.g. Rounce et al. (2017b) and Miles et al. (2018), but the damages are usually limited. Since the pond is very small in your study, it is hard to image it would form a damaging debris flow, but limited socio-economic details are presented.
Response: In the paper (Erokhin et. al 2018) we described in detail the outburst of a small lake (Teztor) by subsurface channels. As a result of the Teztor Lake (50,000 m3) outburst, its volume decreased by only half and still caused a debris flow rate of 300-350 m3/s. That is, the volume of released water was ca. 25,000 m3, a comparable volume with the lake in the Aksay valley. In case of an increase in the lake’s volume, there is a risk of more powerful flows. Moreover, there is a threat of outbursts from an en-glacial pocket that we described in our other paper (Zaginaev et.al, 2016). Many lakes in our catalogue of hazardous lakes have area <20 000m2, similarly to our colleagues’ work in Uzbekistan (Petrov et al, 2017) and Kazakhstan (Blagovechshenskiy et al., 2015). And we use the term ‘lake’ (Erokhin and Zaginaev, 2016). It depends on the region, of course, and the hazardous lakes in Tien Shan are not always comparable in size to those in the Himalayas-Karakoram.
2. There are no details how the debris flow simulation was carried out. What debris flow modelling software was used? What was the outburst hydrograph? What resolution was the DEM? Did the model inundate trekking paths? How long did it take for the lake volume to drain?
Response: For the simulation we used an empirical equation of Herheulidze, it was calculated ‘manually’ for individual cross-sections. We calculated a height of wave and then using cross-sections we demarcated which zones would be affected by debris flows. Then we mapped zones. We do not possess a modelling software (Flo2D, RAMMS, etc) in our Institute in Kyrgyzstan.
It is a difficult question to say how long the lake will drain without knowledge of subsurface channel capacity. This lake could drain within 30 minutes (if outburst flow rate is around 20 m3s-1). We calculated the time when debris flow will reach the lower part of the valley - it is just 8 minutes, velocity of a DF is 8-10 m/s, density of DF more than 2 g/m3 (Erokhin et. al 2018).
3. There is no conclusion. Please add one.
Response: Conclusion was added.
4. The pond has drained during your study e.g. Figure 4d, so why is this drainage, which I assume did not cause a debris flow, likely to become a dangerous event? The volume is very small, so it would only be able to sustain the peak discharge for a very short time.
Response: It happened because channels were not frozen yet and a glacier melting ceased (usually it happens in September). Sometimes a non-stationary lakes drain without any consequences, like Teztor lake in 1988 (Erokhin et al., 2018) and then there was a major event in 2012 when the same lake suddenly drained. We don’t know exactly when and where can subsurface channels open, but sometimes it can be predicted, e.g. based on rising lake water temperature (Erokhin et al., 2018).
Specific comments:
L9. Remove ‘The’.
L14. Perhaps ‘failure of the moraine dam’, or ‘outburst of the proglacial lake’, rather than failure of the lake.
Yes, done.
L14-15. Fieldwork and monitoring are nonspecific. State what was done in the field and monitored.
Done.
L17. State how many people. State specific years.
Response: During summer months people who live in the valley are increasing to 30 (it is staff of National Park), permanently living staff number is ca. 20. We think that this information is not necessary in abstract.
L26. I don’t think it’s true to say lakes with conduits are most hazardous.
Response: In this paper we explain why it is more hazardous for lakes of this type - Erokhin et al., 2018. For instance for lakes landslide dammed type, it is more common to outburst by surface channels.
L33. Please rephrase ‘tectonically active slopes of ranges are very dynamic.’
Done.
L34-35. Add citations to support these statements.
We added citation
L38. GLOFs
L48. Historical archives of what? Please state.
Yes, done.
L60. Figure 1. A, should probably be Figure 1a. Check the journal formatting and correct throughout.
Thank you. Done.
L64. Add ‘Elevation (m a.s.l)’ to the panel (a) legend.
Done.
L83. The catchment area was already stated earlier. Define the ‘high Melton index’.
Done.
L94. Conditions such as what? Please state.
It is mean geomorphological and glaciological conditions. We added information.
L114. Please specify the date.
Done.
L120. In the abstract it is stated after 1980. You need to be specific about the formation and development dates. Add more details in supplementary information if required.
Response: We added a Landsat image and stated the date of formation between 1988 and 1994.
L122. Where is the icefall, please label.
We indicated the icefall on the Fig. 1
L125. Figure 2 doesn’t show when the lake appeared, it could be anywhere from 1988 to 2016. Can you add a Landsat image and be specific about the formation date. State the specific dates for each image.
Response: Thank you for the comment, we found several Landsat images, resolution is very low, but it was possible to recognize a lake on the images of 1994. It means that lake appeared between 1988 and 1994.
L160. Add the equation and r2 value to the trendlines.
Done.
L164. What about Landsat imagery?
Response: There was no convenient Landsat images for this period
L168. Where did this 10 m DEM come from, how was it generated?
Response:We added information to the text. We extracted DEM from Sentinel-1 in SNAP.
L207-208. State the model and uncertainty in the echo sounder used. State what algorithm did you use to interpolate the bathymetry and why?
Response: Thank you for the comment we added this information. We indicate that interpolation algorithm is Kriging, but we don’t think it is necessary to explain why we used this method.
L211. State Agisoft Photoscan and add the version number. How was the DEM built in Agisoft? Did you use ground control points, or the native UAV georeferencing. What was the RMSE?
Response:We did not use ground points. In Agisoft a point cloud was created, then in ArcGIS we created a DEM. RMSE for GPS is 1-2 m. The Phantom 4 Pro (was used for this survey) accuracy is within 0.1 m vertical, and 1.5 m horizontal.
L213. There are currently no details on the modelling.
It was not modelling using special software, it was a calculation based on empirical equation Herheulidze, which we explain in 3.Material and Methods section.
L214. ‘digital elevation model’. How was the digital model combined with the profiles?
Response: Only used on area close to the protective dam. We had a several ground points on the dam in the upper part (cross section 4). We compared a result from drone and results of survey.
L217. ‘The lake…’
L218. How young is very young?
Response: We added more information.
L219. You need to be more specific about the formation date.
Response: we added additional information
L220. ‘falls off’ is not a good term. It is either flowing or calving. Use the degree sign, not ‘0’
Response: term was changed
L221. These are commonly referred to as ‘ice cliffs’ e.g. Watson et al. (2017b),
L223. ‘Figure 4d’.
Response: changed
L238. ‘field work’. Annotate Figure 5 to show where meltwater is entering the lake and ‘the dam’. Replace 0,3 with 0.3 on the legend, and 13,9. Why does the scale bar suggest the lake is approximately 320 m adjacent to the ice cliff in Figure 5, but ~100 m in Figure 4? State the reference elevation of the bathymetric surveys, other the two surveys cannot be compared. The shape of the surveys looks different, suggesting a different water level. So specify the elevations so you can be sure the basin has deepened.
Response: Thank you for this comment. It was a mistake, we corrected it.
L245. Add the July 2012 lake area to Table 2.
Done
L254. State the dates of this area decrease.
L264. Capitalise ‘Uchitel Glacier’ here and throughout. Similar for ‘Uchitel Lake’.
Done
L271. State in cumecs no litres.
Done
L272. The lake is adjacent to the glacier, not ‘under’, which would imply a subglacial lake. Change throughout.
Done
L276. ‘Very hazardous’ isn’t supported by this study without evaluating the mechanisms of how the lake would outburst. Especially if there currently isn’t any water in it. The lake has a very small volume, comparable to supraglacial ponds on Himalayan debris-covered glaciers which drain and reform annually e.g. Miles et al. (2017). Outburst from these ponds can occur, but the volumes are typically much larger, especially to cause any notable impact. See Rounce et al. (2017b).
Response: According to the last inventory of potentially hazardous lakes in Kyrgyzstan, we have around 370 lakes, 80% of them are ice-cored dam type, 20% of the lakes have a volume 20 000 – 30 000 m3. Sometimes we even include to this catalogue an empty depression, it means that such lake is non-stationary and can fill during one or two years and then outburst (Erokhin et. al, 2018). Kyrgyzstan is a mountain country and many villages are located in the mountain valleys, close to the hazards. So even a not so powerfull debris flow (peak discharge 200m3/s during several minutes) can cause significant damage (Shahimardan 1988).
L338. (Figure 7). Please add the glacier outlines to this figure, and perhaps the lake outline. It is not clear otherwise.
Done
L350. What is the ‘end wall’?
Response: We clarified this
L367. Remove –.
Done
L369. There is a difference between the lake benignly draining through conduits and outbursting.
Response: we absolutely agree. We changed it.
L382. Figure 8
L380-399. There are no details in the methods of how this simulation was carried out. What software? Was the input hydrograph? How was the debris entrainment determined?
Response: Using empirical equation of Herheulidze we calculated the height of outburst wave and then from cross section to cross section we manually check which zones will be affected and these zones were drawn on the map for each scenario.
L404 Table 3. Replace commas with ‘.’
Done
L407. National Park.
Done
L409. If you are stating a GLOF is a danger, you should indicate a probability of occurrence. How many visitors visit annually?
Response: We posed this question to the Ala-Archa National Park. They say that in June, July and August, on average there are 500 visitors per day. But we did not find any official data for reference.
L410. Does your simulation reach this city? It is not shown. Where would the large GLOF event originate given the lake is small?
Response: No, it means that it can influence the discharge but not in case this lake outbursts (it can be just peak discharge around 300 m3/s). It was in the past in the Aksay valley in 1968 and 1969 and it refers to threat from Aksay Glacier for major events.
L427. Be specify about the lake formation.
done
L445. Underground is probably incorrect. I assume you mean through ice, or englacial.
Response: ‘Englacial’ is used for just pure ice without moraine deposits, in our case, there is morainic material with buried ice blocks/lenses.
L452. Restructuring.
Done
L454-457. This is just speculation and should be removed or supported with evidence from your study.
Response: we agree with reviewer we added a reference.
L470. Evacuation of who? The trekkers?
Response: Evacuation of tourists as well as the staff of the National Park.
References
Zaginaev, V.; Ballesteros-Cánovas, J.; Matov, E.; Petrakov, D.; Stoffel, M. Reconstruction of glacial lake outburst floods in northern Tien-Shan: implications for hazard assessment. Geomorphology 2016, 269, 75–84. doi:10.1016/j.geomorph.2016.06.028
Erokhin, S.; Zaginaev, V.; Meleshko, A.; Ruiz-Villanueva, V. et al. Debris flows triggered from non-stationary glacier lake outbursts: the case of the Teztor Lake complex (Northern Tian Shan, Kyrgyzstan). Landslides 2018, 1-16. doi:10.1007/s10346-017-0862-3
Petrov, M. A., Sabitov, T. Y., Tomashevskaya, I. G., Glazirin, G. E., Chernomorets, S. S., Savernyuk, E. A., ... & Mountrakis, G. (2017). Glacial lake inventory and lake outburst potential in Uzbekistan. Science of the Total Environment, 592, 228-242.
Viktor Blagovechshenskiy, Vasiliy Kapitsa and Nikolay Kasatkin (2015). Danger of GLOFs in the Mountain Areas of Kazakhstan Journal of Earth Science and Engineering 5, 182-187 doi: 10.17265/2159-581X/2015. 03. 003
Erokhin, S.; Zaginaev V. The forecast of a outburst probability of the mountain lakes of Kyrgyzstan on to basis of their catalog, 13th edn. Ministry of Emergency Situations of the Kyrgyz Republic, Bishkek. 2016, 627–639.
References
Miles, E.S. Steiner, J. Willis, I. Buri, P. Immerzeel, W.W. Chesnokova, A. and Pellicciotti, F. 2017. Pond Dynamics and Supraglacial-Englacial Connectivity on Debris-Covered Lirung Glacier, Nepal. Frontiers in Earth Science. 5(69). http://dx.doi.org/10.3389/feart.2017.00069.
Rounce, D.R. Byers, A.C. Byers, E.A. and McKinney, D.C. 2017b. Brief communication: Observations of a glacier outburst flood from Lhotse Glacier, Everest area, Nepal. The Cryosphere. 11(1), 443-449. 10.5194/tc-11-443-2017.
Watson, C.S. Quincey, D.J. Carrivick, J.L. Smith, M.W. Rowan, A.V. and Richardson, R. 2017a. Heterogeneous water storage and thermal regime of supraglacial ponds on debris-covered glaciers. Earth Surface Processes and Landforms. 229-241. http://dx.doi.org/10.1002/esp.4236.
Watson, C.S. Quincey, D.J. Smith, M.W. Carrivick, J.L. Rowan, A.V. and James, M. 2017b. Quantifying ice cliff evolution with multi-temporal point clouds on the debris-covered Khumbu Glacier, Nepal. Journal of Glaciology. 823-837. http://dx.doi.org/10.1017/jog.2017.47.

Reviewer 3 Report
Review to the manuscript ‘Development of a potentially hazardous pro-glacial Aksay lake, Kyrgyz range, northern Tien Shan‘ submitted by V.Zaginaev et al. to Hydrology.
Presented case study deals with the evolution of recently formed glacial lake Aksay in northern Tien Shan. In my opinion, this manuscript in its current form does not meet the parameters of scientific article. Firstly, only very little original / novel data are presented (apart from two bathymetrical surveys and basic analysis of remotely sensed images). The manuscript suffers from the excessive amount of general or even trivial and descriptive statements which are often repeating and have only little value for readers. More sophisticated analysis of remotely sensed data in combination with field observations might provide additional and novel insights. Secondly, modelling part is weak, considering recent progress in this field. The scenarios used (300 - 900 m3/s) seem unrealistic, considering lake volume of about 27,000 m3 and suggested outburst mechanism (opening of underground outflow channels). I’m convinced that in case of outburst, the process of opening channels would be rather graduall, considering max. depth of 14 m (low hydrostatic pressure) and peak discharge, thus, much lower. I suggest to the authors to check within the literature, what is typical duration of this outburst mechanism, to come up with realistic estimation of peak discharge from the lake of this type and volume. Thirdly, any claims on hazards posed by this lake (e.g, L368: ‘chances are high‘), thus, seem to me more like a sensation hunting, if not supported by rigorous research. I’m convinced that giving more emphasis on the evolution of the entire valley, building on repeated field surveys the authors did, might be of certain interest for readers rather than shallow searching for hazards. Fourthly, the overall quality of the English within the manuscript is poor, making parts of it hardly understandable. I suggest proofreading by native speaker.
To conclude, I’m sorry to say that the manuscript seems more like a preliminary study and is not acceptable for scientific journal in its current form. Therefore I suggest substantial reworking and resubmission. Hope that my comments and suggestions will help to revise the manuscript.
Author Response
Reviewer #3
Review to the manuscript ‘Development of a potentially hazardous pro-glacial Aksay lake, Kyrgyz range, northern Tien Shan‘ submitted by V.Zaginaev et al. to Hydrology.
Presented case study deals with the evolution of recently formed glacial lake Aksay in northern Tien Shan. In my opinion, this manuscript in its current form does not meet the parameters of scientific article. Firstly, only very little original / novel data are presented (apart from two bathymetrical surveys and basic analysis of remotely sensed images). The manuscript suffers from the excessive amount of general or even trivial and descriptive statements which are often repeating and have only little value for readers. More sophisticated analysis of remotely sensed data in combination with field observations might provide additional and novel insights. Secondly, modelling part is weak, considering recent progress in this field. The scenarios used (300 - 900 m3/s) seem unrealistic, considering lake volume of about 27,000 m3 and suggested outburst mechanism (opening of underground outflow channels). I’m convinced that in case of outburst, the process of opening channels would be rather graduall, considering max. depth of 14 m (low hydrostatic pressure) and peak discharge, thus, much lower. I suggest to the authors to check within the literature, what is typical duration of this outburst mechanism, to come up with realistic estimation of peak discharge from the lake of this type and volume. Thirdly, any claims on hazards posed by this lake (e.g, L368: ‘chances are high‘), thus, seem to me more like a sensation hunting, if not supported by rigorous research. I’m convinced that giving more emphasis on the evolution of the entire valley, building on repeated field surveys the authors did, might be of certain interest for readers rather than shallow searching for hazards. Fourthly, the overall quality of the English within the manuscript is poor, making parts of it hardly understandable. I suggest proofreading by native speaker.
To conclude, I’m sorry to say that the manuscript seems more like a preliminary study and is not acceptable for scientific journal in its current form. Therefore I suggest substantial reworking and resubmission. Hope that my comments and suggestions will help to revise the manuscript.
Response: We thank the reviewer for the time and effort dedicated to the revision of our work. We worked on the original version and tried to improve it so that it does not resemble a preliminary study. We explain our motivation for this study below.
This scientific article is a synthesis of several years of work and includes the results of a decade of work on monitoring of the lake and the valley. In addition to the bathymetric measurements, we compiled a geomorphological map based on field work, and profiling of the valley was carried out thanks to which 26 detailed longitudinal cross sections were constructed.
In the paper (Erokhin et. al 2018) we described in detail the outburst of the lake with a small volume by subsurface channels. As a result of Teztor’s Lake outburst (50,000 m3), its volume decrease by only half and caused a debris flow rate of 300-350 m3 /s. That is, outbursted volume was ca. 25,000 m3, a comparable volume to the lake in the Aksay valley. In case of an increase in the lake’s volume, there is a risk of higher flow rate events. Also, there is a potential for outbursts from the intra-glacial pocket that we described in our other paper (Zaginaev et.al, 2016).About 90% of the registered outbursts that caused damage in Kyrgyzstan (70 cases recorded in the last 60 years) occurred after outbursts by subsurface channels. This is the most common type of outbursts in the region, and in very rare cases the outburst occurred in a combined way - the dam collapsed during the outburst (Shakhimardan, 1998).We witnessed an outburst of Chelektor Lake in 2017 (northern Tien Shan) in the lower part of the valley and then we visited a lake in 2017 and 2018. Volume of the lake before outburst was 75,000 m3, the lake drained only by half and caused a debris flow of peak discharge of 400 m3/s. The lake outbursted simultaneously through three channels and it was not a gradual breach for several hours, the breach lasted only 20-30 minutes. In each case, the outburst time depends on the size and number of channels.
The drainage channels of this pro-glacial lake are located almost at the bottom of the lake, that is, the lake would probably lose almost the entire volume. After our paper in 2016, the dam was not reinforced well enough. Flows in July 2017 with a flow rate of 100 m3/s caused by heavy rains and intensive melting damaged the dam. We choose a three scenarios (300,600 and 900 m3 /s) for future based on historical data (Zaginev et al, 2016) and our experience. In our article we mentioned about threat from Aksay Glacier. In the past during 10 years were 10 events after outburst from englacial water tanks, two of them (1968 and 1969) were catastrophic with peak discharge 900 m3 /s). We mentioned in our article that it can be repeat.
This scientific article aims to draw attention and help to protect people visiting this very popular place (in addition to hotels in the lower part of the valley, there is the residence of the President of Kyrgyzstan). We do not do science just for the sake of science, we are not looking for sensations. This is a concrete example of how our research is actually applied in practice. Scientific articles are a means of communication not only for scientists among themselves. In addition to them, there are people who make decisions (e.g. from national institutes) and who are interested in such studies.
References
Zaginaev, V.; Ballesteros-Cánovas, J.; Matov, E.; Petrakov, D.; Stoffel, M. Reconstruction of glacial lake outburst floods in northern Tien-Shan: implications for hazard assessment. Geomorphology 2016, 269, 75–84. doi:10.1016/j.geomorph.2016.06.028
Erokhin, S.; Zaginaev, V.; Meleshko, A.; Ruiz-Villanueva, V. et al. Debris flows triggered from non-stationary glacier lake outbursts: the case of the Teztor Lake complex (Northern Tian Shan, Kyrgyzstan). Landslides 2018, 1-16. doi:10.1007/s10346-017-0862-3

Round 2
Reviewer 3 Report
I thank the authors for their reply and edits. I’m glad to see that they made certain progress with the manuscript. Still, I’d like to clarify some issues with the authors. I insert my comments (in blue) to the original reply of the authors (in red):
- - -
Response: We thank the reviewer for the time and effort dedicated to the revision of our work. We worked on the original version and tried to improve it so that it does not resemble a preliminary study. We explain our motivation for this study below.
This scientific article is a synthesis of several years of work and includes the results of a decade of work on monitoring of the lake and the valley. In addition to the bathymetric measurements, we compiled a geomorphological map based on field work, and profiling of the valley was carried out thanks to which 26 detailed longitudinal cross sections were constructed.
- if I understand it correctly, the modelling approach used by the authors does expect constant peak discharge thorough the valley (P1-P20; Tab. 3) - if yes, please clarify this in methodology (I also suggest to discuss this modelling approach; see below)
In the paper (Erokhin et. al 2018) we described in detail the outburst of the lake with a small volume by subsurface channels. As a result of Teztor’s Lake outburst (50,000 m3), its volume decrease by only half and caused a debris flow rate of 300-350 m3 /s. That is, outbursted volume was ca. 25,000 m3, a comparable volume to the lake in the Aksay valley.
- this is exactly the point; lake outburst usually does not involve entire lake volume, but (often small) part of it, because hydrostatic pressure decreases with decreasing lake level (“to-bottom” lake outbursts are, actually, extremely rare and exclusively caused by high magnitude triggers (e.g. ice avalanche) which are not expected in case of studied lake);
- Teztor Lake outburst is good example - applying the fact that about a half of retained water was released to the studied lake in front of Uchitel glacier, an outburst of 13,500 m3 might be expected
- anyway, even considering the release of 27,000 m3 (I’d call this “the worst case scenario”) expected peak discharge could be somewhat similar to 300-350 m3/s; not 600 m3/s nor 900 m3/s
In case of an increase in the lake’s volume, there is a risk of higher flow rate events. Also, there is a potential for outbursts from the intra-glacial pocket that we described in our other paper (Zaginaev et.al, 2016).- it is not clear from the manuscript if the authors consider also future lake growth or the possibility of lake outburst from intra-glacial lake (?); if so, this needs to be clearly stated throughout the manuscript
About 90% of the registered outbursts that caused damage in Kyrgyzstan (70 cases recorded in the last 60 years) occurred after outbursts by subsurface channels. This is the most common type of outbursts in the region, and in very rare cases the outburst occurred in a combined way - the dam collapsed during the outburst (Shakhimardan, 1998).We witnessed an outburst of Chelektor Lake in 2017 (northern Tien Shan) in the lower part of the valley and then we visited a lake in 2017 and 2018. Volume of the lake before outburst was 75,000 m3, the lake drained only by half and caused a debris flow of peak discharge of 400 m3/s. The lake outbursted simultaneously through three channels and it was not a gradual breach for several hours, the breach lasted only 20-30 minutes.
- again, considering the release of about 37,500 m3 and observed peak discharge 400 m3/s, expected peak discharge for the lake with the total volume of 27,000 m3 would be lower, especially considering partial lake drainage
In each case, the outburst time depends on the size and number of channels.
The drainage channels of this pro-glacial lake are located almost at the bottom of the lake, that is, the lake would probably lose almost the entire volume. - this is interesting information; I need the authors to further explain how do they know that drainage channels are located almost at the bottom of the lake
After our paper in 2016, the dam was not reinforced well enough. Flows in July 2017 with a flow rate of 100 m3/s caused by heavy rains and intensive melting damaged the dam. We choose a three scenarios (300,600 and 900 m3 /s) for future based on historical data (Zaginev et al, 2016) and our experience.
In our article we mentioned about threat from Aksay Glacier. In the past during 10 years were 10 events after outburst from englacial water tanks, two of them (1968 and 1969) were catastrophic with peak discharge 900 m3 /s). We mentioned in our article that it can be repeat.
- in my opinion, 900 m3/s recorded from Aksay glacier can’t be simply applied as “the worst case scenario” for the 27,000 m3 proglacial lake in front of the Uchitel glacier, despite geographical proximity
This scientific article aims to draw attention and help to protect people visiting this very popular place (in addition to hotels in the lower part of the valley, there is the residence of the President of Kyrgyzstan). We do not do science just for the sake of science, we are not looking for sensations. This is a concrete example of how our research is actually applied in practice. Scientific articles are a means of communication not only for scientists among themselves. In addition to them, there are people who make decisions (e.g. from national institutes) and who are interested in such studies.- this is obvious; my point is that in hazard modelling (in general), only realistic scenarios must be used, especially when focusing on decision-makers;
- I recommend to the authors to see and refer to the GAPHAZ guideline: http://gaphaz.org/files/Assessment_Glacier_Permafrost_Hazards_Mountain_Regions.pdf
References
Zaginaev, V.; Ballesteros-Cánovas, J.; Matov, E.; Petrakov, D.; Stoffel, M. Reconstruction of glacial lake outburst floods in northern Tien-Shan: implications for hazard assessment. Geomorphology 2016, 269, 75–84. doi:10.1016/j.geomorph.2016.06.028
Erokhin, S.; Zaginaev, V.; Meleshko, A.; Ruiz-Villanueva, V. et al. Debris flows triggered from non-stationary glacier lake outbursts: the case of the Teztor Lake complex (Northern Tian Shan, Kyrgyzstan). Landslides 2018, 1-16. doi:10.1007/s10346-017-0862-3
To conclude, I need the authors to further justify or omit 900 m3/s (600 m3/s) scenarios (I, honestly, prefer to omit). Based on the examples stated above, I disagree that it could originate from the studied 27,000 m3 proglacial lake located in front of Uchitel glacier. I suggest not to include this scenario within the manuscript if focusing mainly on this lake. If the authors wish to keep this scenario, I strongly suggest to stress that this is considering hypothetical drainage of en-/intra- glacial lake (evidence of such lake would be, however, still missing).
Apart from that, I suggest to include field photos from the supplement in the main body of the manuscript (in my opinion, these might be of interest for readers). Finally, I would welcome some discussion on modelling approach used (what are advantages / uncertainties?). Some of the newly introduced conclusions are not supported by results (especially these related to the damage on artificial dam) and should be reformulated or moved to discussion. Until these issues are addressed, I can’t recommend the manuscript for the publication, therefore I suggest some additional major revisions.
Some specific comments:
L301: please check this number
L375-380: please add reference(s)
L703-705: see my general comments above
L1065: this is actually quite far away; do you expect GLOF as a result of opening this 500 m long underground channel? Please discuss
L1066-1067: have you measured also the inflow into the lake?
Fig. 6: please add information about the topography (contour lines)
L1518-1525: this is rather a discussion, not supported by rigorous modelling; moreover, 600 m3/s and 900 m3/s scenarios from 27,000 m3 proglacial lake in front of Uchitel valley is not realistic considering above mentioned examples of past GLOFsL1979-1981: this glacier retreat-induced lake growth is not a subject of this study; this statement is not supported by data / results and should not be mentioned here
L1982-1984: this statement about artificial dam failure is not based on the results of the study, better omit or move to the discussion
References: apart from own work of the authors, the list of references is bit outdated and recent literature is ignored; please check for updates on recent literature (2015-2018) in both GLOF research as well as modelling
